# Position: Weight Space Should Be a First-Class Generative AI Modality

**Zhangyang Wang** [1]   **Peihao Wang** [1]   **Kai Wang** [2]

## Abstract

Neural network checkpoints have quietly become a large-scale data resource: millions of trained weight vectors now exist, each encoding task-, domain-, and architecture-specific knowledge. This position paper argues that *model checkpoints should be treated as a first-class data modality, and that generative modeling in weight space should be standardized as a core machine learning primitive*. Recent advances demonstrate that neural weights can be synthesized on demand, often matching fine-tuning performance while reducing adaptation cost by orders of magnitude. We contend that these results reflect an underlying structural fact: high-performing models occupy low-dimensional, highly structured regions of weight space shaped by symmetry, flatness, modularity, and shared subspaces. Building on this view, we organize existing methods into a five-stage pipeline, survey applications where the approach is already practical, and clarify current limits: adapter-scale and conditional generation are advancing rapidly, while unrestricted frontier-scale checkpoint synthesis remains open. Our goal is to shift the community's default mindset from optimizing models per task to sampling models from learned weight distributions, accelerating toward an era in which *AI systems routinely improve or create other AI systems*.

## 1. Introduction and Statement of Position

*The success of generative AI has so far been asymmetric: we can* generate *pixels, waveforms, and tokens on demand, but we still* construct - *via repeated, costly optimization - the models that perform this generation.* This asymmetry increasingly defines the bottleneck of modern machine learning. Training a single foundation model can emit as much $CO_2$ as a small airline fleet (Patterson et al., 2021),

while even modest fine-tuning remains prohibitive for many start-ups, academic groups, and edge deployments. At the same time, demand for *ultra-rapid*, *task-specialized*, and *user-personalized* models is accelerating—from on-device copilots to adaptive robotics to personalized healthcare.

We argue that progress is throttled less by hardware than by *methodological inertia*. Neural network parameters are still treated as immutable artifacts to be rediscovered from scratch for every new task or user. Yet a century of statistics teaches a different lesson: when data exhibit structure, we should *model and sample* that structure rather than repeatedly re-optimizing it. Recent results indicate that trained neural weights themselves form such a structured data modality. HyperNetworks generate full convolutional weight tensors in a single forward pass (Ha et al., 2017); graph-conditioned predictors initialize unseen architectures without task-specific training (Knyazev et al., 2021); diffusion models denoise checkpoints in weight space to yield ImageNet-ready ConvNeXt backbones (Wang et al., 2024; 2025). More recently, (Liang et al., 2025) demonstrates that even large language models (LLMs) can be adapted by *directly generating* low-rank weight updates in seconds, often matching or exceeding fine-tuning performance.

**Position.**   We posit that ***neural network checkpoints constitute a first-class data modality, and that generative modeling in weight space should be standardized as a core machine learning primitive.*** Under this view, model creation is reframed as *conditional sampling*: given a task description, data domain, user context, or architectural specification, one samples from a learned distribution of high-performing weights. Crucially, this position does *not* claim that weight generation solves all problems of training or scalability. Nor is it reducible to the connectedness of minima (Entezari et al., 2022; Garipov et al., 2018): connectivity guarantees that paths exist, but it does not explain how to reliably reach flat, robust, calibrated, or domain-adapted regions. Achieving those properties requires learning an explicit, conditioned density over weight configurations.

**Why Now?**   Weight-space generation is no longer a speculative idea; it is enabled by a concrete shift in the ML ecosystem. Open-science efforts have produced an unprecedented corpus of trained models: (i) the Hugging Face Hub alone hosts over one million public checkpoints across modalities; (ii) TensorFlow Hub, ONNX Model Zoo, and open releases

---

[1]University of Texas at Austin [2]Tencent Hy. Correspondence to: Zhangyang "Atlas" Wang <atlaswang@utexas.edu>.

*Proceedings of the 43rd International Conference on Machine Learning*, Seoul, South Korea. PMLR 306, 2026. Copyright 2026 by the author(s).

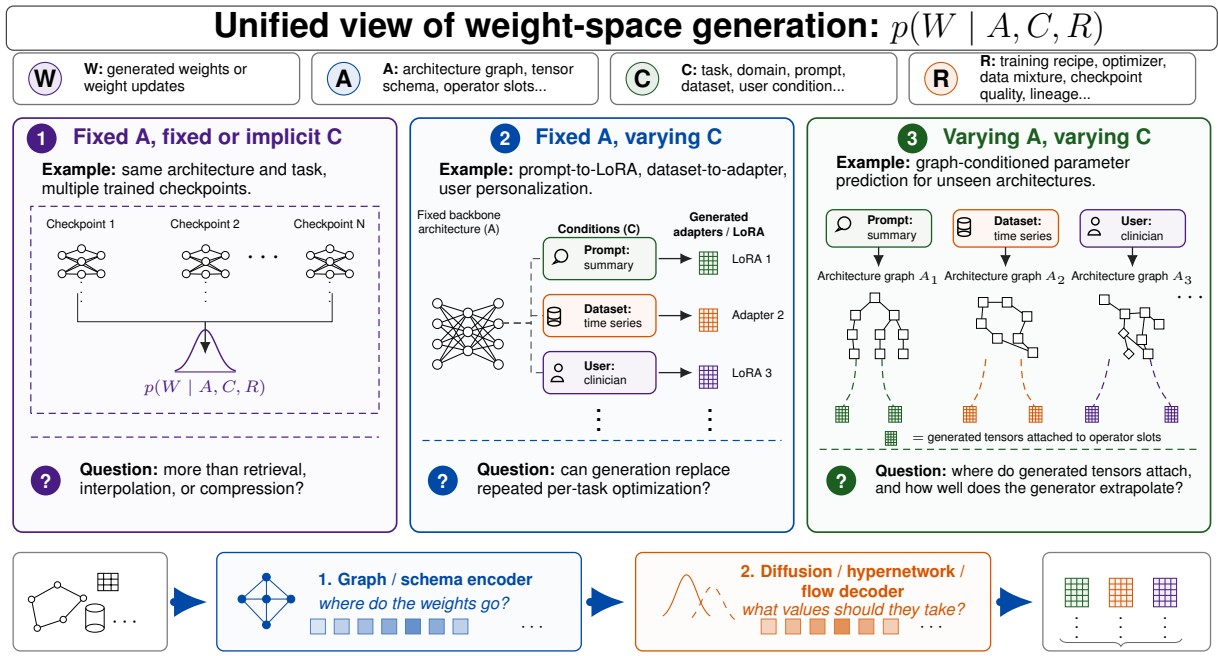

*Figure 1.* A regime map for weight-space generation (see §3). We frame neural weight generation as conditional sampling from $p(W \mid A, C, R)$, where $W$ denotes generated weights or weight updates, $A$ specifies the architecture graph and tensor schema, $C$ encodes task or user conditions, and $R$ captures training-recipe and checkpoint-lineage information. The three regimes distinguish whether architecture and conditioning are fixed or varied: checkpoint-level generation under fixed settings, condition-driven generation for adapters or personalization, and graph-conditioned generation across architectures. This view separates the placement problem, handled naturally by graph or schema encoders, from the value-generation problem, handled by diffusion, hypernetwork, or flow decoders.

from major industrial labs further expand this repository; (iii) recent academic studies explicitly treat neural weights as data, including large-scale analyses of tens of thousands of trained models; (iv) parameter-efficient fine-tuning has exploded the number of *task-conditioned* checkpoints, with over 22 000 LoRA adapters for the LLaMA family alone as of 2025. Together, these trends provide both the *data substrate* and the *deployment pressure* that make generative modeling in weight space timely and actionable.

**Contributions.** This paper advances the position through four concrete contributions:

1. **Theory.** We synthesize decades of optimization theory and empirical results, to explain why generative modeling of weights is both feasible and non-trivial.

2. **Framework.** We formulate weight-space generation as conditional sampling $p(W \mid A, C, R)$ (Fig 1), distinguishing fixed-architecture generation, condition-driven adaptation, and cross-architecture generation, and relate these regimes to a five-stage pipeline: tokenization $\rightarrow$ embedding $\rightarrow$ generative predictor $\rightarrow$ training strategy $\rightarrow$ evaluation.

3. **Evidence.** We survey demonstrated applications, while separating practical current regimes from still-open frontier-scale full checkpoint generation.

4. **Agenda.** We analyze alternative views, failure modes, and concrete steps toward this new paradigm.

By elevating weights from optimization artifacts to generative objects, we aim to shift the community's default question from *"How do we fine-tune this model?"* to *"What distribution of models should we sample from?"*

## 2. Theoretical Foundation of Weight Space Geometry: Why Feasible, Yet Non-Trivial

If neural network weights are to be treated as a *generative modality*, then any successful generator must respect the intrinsic geometry of weight space itself. This section synthesizes what decades of optimization theory and recent empirical studies reveal about that geometry, and explains how they both *enable* and *constrain* weight generation.

*Far from forming a random high-dimensional cloud*, the set of high-performing neural networks concentrates on structured subsets of parameter space. These structures explain why learned generative models can empirically succeed.

### 2.1. Mode Connectivity of Solutions

Partly because of over-parameterization, many trained neural networks do not behave as isolated optima in the raw parameter space. While naive linear interpolation between independently trained models often incurs large loss barri-

ers (Shevchenko & Mondelli, 2020; Kuditipudi et al., 2019), empirical results revealed *mode connectivity*: continuous low-loss paths linking solutions trained from different initializations (Garipov et al., 2018; Draxler et al., 2018).

Theoretical work clarifies when such connectivity arises. Solutions exhibiting dropout or noise stability admit barrier-free connections in ReLU networks (Kuditipudi et al., 2019), and sufficiently wide networks provably admit near-zero-loss paths between SGD solutions (Shevchenko & Mondelli, 2020). Even moderately over-parameterized architectures, such as CIFAR-10 ResNets, empirically exhibit this phenomenon (Garipov et al., 2018; Draxler et al., 2018).

Connectivity provides evidence that many low-training-loss SGD solutions are not isolated in parameter space. It does not by itself prove that all well-generalizing solutions form a single manifold. Furthermore, this result alone does *not* make weight generation trivial. Connectivity guarantees that *paths* exist, but offers no mechanism for *sampling* points with desired properties such as robustness, calibration, or transferability. For generation, the challenge is hence not reachability, but *density modeling* (see $2.4): reachability between trained points is often possible, but sampling robust, calibrated, transferable, or safe models still requires learning a structured conditional density.

## 2.2. Permutation Symmetries and Quotient Geometry

A central reason for the apparent multiplicity of minima is parameter symmetry (Goodfellow et al., 2015; Choromanska et al., 2015). Permuting hidden units or channels preserves the network's function while producing numerically distinct weight vectors. As a result, each functional solution corresponds to an equivalence class in raw parameter space, creating flat directions and degenerate minima.

Interpolation between unaligned solutions introduces artificial loss barriers due to mismatched neuron identities. When permutation symmetry is explicitly factored out, these barriers vanish. (Entezari et al., 2022) conjectured that, modulo permutation, SGD solutions occupy a single basin. *Git Re-Basin* (Ainsworth et al., 2023) provided constructive evidence, demonstrating zero-barrier linear connectivity after neuron alignment. Related symmetry-aware matching improves model averaging and federated learning (Yurochkin et al., 2019; Tatro et al., 2020; Zhao et al., 2024b).

For generative modeling, this implies that the natural object of study is not raw weight space $\mathbb{R}^n$, but a *quotient space* under symmetry. Generators that ignore this structure waste capacity modeling redundant degrees of freedom.

## 2.3. Flatness and Low Intrinsic Dimension

Beyond symmetry, trained solutions occupy remarkably flat regions of the loss landscape. Hessian analyses show that only a small number of eigenvalues are large at SGD min-

ima, with most directions exhibiting near-zero curvature (Sagun et al., 2017; Ghorbani et al., 2019). Perturbations along flat directions often leave performance unchanged. Flatness correlates strongly with generalization (Jiang et al., 2020), and invariant measures confirm that broader basins generalize better despite reparameterization issues (Dinh et al., 2017; Petzka et al., 2021). Methods that explicitly promote flatness, such as Sharpness-Aware Minimization (Foret et al., 2021), consistently improve performance.

Multiple lines of evidence indicate that the *effective* dimension of optimization trajectories and searchable subspaces can be far smaller than the ambient parameter count. Lottery Ticket and subspace training results show that networks can be trained successfully within low-dimensional affine subspaces (Frankle & Carbin, 2019; Chen et al., 2020; Li et al., 2018; Jaiswal et al., 2025). Large-scale training dynamics reveal that SGD trajectories explore a tiny, low-rank subspace of parameter space regardless of initialization (Gur-Ari et al., 2018; Mao et al., 2024), though not without caveat whether a single dominant subspace really exists (Song et al., 2025). Recent memory-efficient optimizer designs (Zhao et al., 2024a; Zhu et al., 2025) explicitly exploit this phenomenon: see (Balzano et al., 2025) for more reviews. These results imply that high-performing weights occupy a *thin, structured manifold*. Sampling must therefore capture strong inter-parameter correlations.

## 2.4. Implicit Bias: Why Solution Density Is Not Uniform

Optimizers exhibit strong implicit bias toward solutions with specific geometric properties. This is critical for weight generation: the target distribution is *highly non-uniform*. Effective generators must learn to model this density.

In deep linear and homogeneous networks, gradient descent with weight decay provably minimizes effective rank over training trajectories (Ji & Telgarsky, 2019; Le & Jegelka, 2022). More broadly, SGD combined with common regularizers implicitly performs rank minimization (Galanti et al., 2024), explaining the low effective rank observed. When data lie on a low-dimensional manifold, optimal solutions align with that intrinsic structure (Ongie et al., 2020).

As another example, SGD also favors small-norm, max-margin solutions in classification settings (Soudry et al., 2018; Lyu & Li, 2020). Although theory for nonlinear networks remains incomplete, empirical evidence consistently shows that optimization concentrates probability mass in a small subset of geometrically regular solutions.

## 2.5. Compositionality and Modularity

Neural networks frequently internalize compositional structure when trained on structured data. Empirical studies show that modern networks decompose into coherent sub-circuits corresponding to reusable functions or concepts (Csordás

et al., 2021; Qiu et al., 2024; Zhang et al., 2024), with individual blocks exhibiting interpretable roles (Bau et al., 2020). Recent theoretical and empirical work suggests that gradient-based training induces latent modular representations within weight space (Cranmer et al., 2020; Zheng et al., 2022; Chen et al., 2023; Wang & Wang, 2025). Network morphisms such as Net2Net (Chen et al., 2016) and progressive stacking (Gong et al., 2019; Wang et al., 2023b;c; Yang et al., 2022) explicitly exploit this modularity. Model editing and merging (Wortsman et al., 2022; Zhao et al., 2024b) further demonstrate empirically that specific skills correspond to localized parameter subsets.

For generation, modularity explains both promise and difficulty. On one hand, generators can recombine learned modules to generalize across tasks. On the other, module boundaries are not smooth or spatially local in raw parameter coordinates, complicating naive notions of locality and requiring structured tokenization and curriculum strategies.

### 2.6. Shared Representations and Universal Subspaces?

Despite architectural diversity and/or training data or task differences, trained networks often converge toward remarkably similar internal representations. The *Platonic representation hypothesis* (Huh et al., 2024) was the first to posit that high-performing models recover a common structure dictated by the data-generating process rather than architectural details. More evidence includes linear alignment of representations across models (Kornblith et al., 2019) snd successful model stitching (Bansal et al., 2021). Recent large-scale empirical studies suggest the existence of shared low-dimensional subspaces that capture variation across datasets, initializations, tasks and architectures (Kaushik et al., 2025). Although still preliminary, this discovery has the potential to explain the success of transfer learning, distillation, and adapter-based methods. It also suggests that generators potentially need not learn architecture-specific distributions from scratch, but can operate in shared latent coordinates that instantiate across architectures.

**Summary.** Taken together, neural weight space is structured and low-dimensional, biased by optimization, making weight generation *feasible*. At the same time, the absence of smooth locality, the presence of symmetry, and the non-uniform density make the problem fundamentally non-trivial. Any successful generator must therefore be symmetry-aware, bias-aligned, and capable of capturing long-range correlations rather than naive smoothness.

## 3. Practical Mechanisms for Neural Weight Generation: A Standardized Taxonomy

Armed with the geometric insights of Section 2, we now turn to the practical question: *how can we learn a model that samples from the structured and low-dimensional distri-*

*bution of high-performing neural weights?* Contemporary systems increasingly converge on a modular pipeline that mirrors the workflow of modern generative AI for images and text, but with crucial differences: weights exhibit strong permutation symmetries, weak "spatial locality" in raw coordinates, and heavy-tailed inter-layer dependencies. As a result, each stage of the pipeline must be re-designed to encode the right invariances and to avoid degenerate solutions.

In this section, we synthesize the rapidly growing literature into a standardized, five-stage blueprint: **tokenization**, **embedding**, **generative predictor**, **training strategy**, and **evaluation**. Our goal is *mechanistic*: to explain how each design choice operationalizes the geometry of Section 2, and to highlight which choices appear most practical today.

**Regimes and scope.** To avoid overstating feasibility, we separate three regimes that recur throughout this section: (i) *adapter generation* (e.g., LoRA-like low-rank updates), (ii) *mid-scale full-weight generation* (e.g., 10M–200M parameter vision backbones), and (iii) *cross-architecture generalization* (predicting weights for unseen graphs). The field is currently most mature in (i) and increasingly competitive in (ii), while (iii) remains an open frontier.

As depicted in Fig. 1, a useful unifying abstraction is the conditional family $p(W \mid A, C, R)$, where $W$ denotes weights or weight updates, $A$ denotes the architecture graph and tensor schema, $C$ denotes task, domain, prompt, dataset, or user condition, and $R$ denotes training recipe and quality metadata[1]. When $A$ is fixed, the parameter slots are known and the generator mainly models values. When $C$ varies, the generator performs conditional adaptation over a known layout. When $A$ varies, the generator must also map generated tensors back to operator types, shapes, and connectivity. In this sense, graph encoders mainly solve the placement problem, while diffusion, flow, or hypernetwork modules solve the value-generation problem.

### 3.1. Tokenization of Weights: What Is "Local"?

Tokenization maps heterogeneous parameter tensors into a common "language" suitable for sequence, set, or graph models. Yet one must confront a key concern: *weight space lacks obvious locality*. Unlike pixels or tokens, raw weights do not live on a natural grid. Instead, locality is *architectural*: weights incident to the same neuron/channel are coupled; kernels share spatial structure; attention blocks couple across layers through residual streams; and symmetries introduce many equivalent coordinate representations.

We therefore view tokenization as choosing an *inductive bias for dependency structure*. Three families dominate

---

[1]In practice, $R$ need not be a user-facing input: a user may specify only $C$, while the generator marginalizes over, selects, or internally conditions on suitable recipe and quality factors.

practice: see next. We believe the practical trend is toward *hybrid locality*: preserve intra-layer structure via chunking and tags, while learning inter-layer dependencies via sequence models, recurrence, or diffusion refinements.

**(1) Flattening and chunking (sequence view).** A naive approach flattens all parameters into one long vector, but this destroys layer structure and aggravates symmetry. Early hypernetwork-based NAS (e.g., SMASH (Brock et al., 2018)) used coarse flattening, while subsequent work emphasized structured partitions and function-preserving transformations such as Net2Net (Chen et al., 2016).

Modern large-scale generators partition weights into coherent tokens. RPG (Wang et al., 2025) slices each layer into fixed-length chunks, applies per-layer normalization, and appends positional tags (layer index, chunk index) to break permutation ambiguity across tokens. SANE similarly operates on thousands of tokens, but encodes them into learned embeddings rather than directly modeling raw floats (Schürholt et al., 2024). This "token-with-position" paradigm implements a weak form of locality: it assumes that chunks within a layer are more correlated than chunks across layers, while still permitting the generator to learn global dependencies through attention or recurrence.

**(2) Set encodings (permutation-invariant view).** Permutation symmetry (§ 2) suggests representing collections of neurons/channels as *unordered sets*. Set encoders and permutation-invariant pooling attempt to quotient out label symmetries at the representation level. For example, Set-Based Neural Network Encoding (Andreis et al., 2024) treats layer parameters as sets of neuron vectors and uses permutation-invariant operations to form representations. Such tokenization can reduce spurious variation, but may fail to model long-range inter-layer correlations.

A practical lesson from *Git Re-Basin* (Ainsworth et al., 2023) and related alignment work is that *symmetry can be handled either before modeling (canonicalization / alignment) or inside the model (permutation-invariant encoders)*. Most current generators choose the latter implicitly (via tags, invariances, and data augmentation), because perfect canonicalization is itself a hard problem.

**(3) Graph tokenization (architecture-as-graph view).** When architectures vary, tokenization naturally becomes graph-structured. Graph HyperNetworks (Knyazev et al., 2021) treat the network as a DAG $\mathcal{G} = (\mathcal{V}, \mathcal{E})$, with node attributes encoding layer types and shapes; message passing propagates context and produces a weight prediction per node. Graph MetaNetworks (Lim et al., 2024) extend this idea to diverse architectures, aiming to predict parameters for unseen graphs. Graph tokenization is the clearest path toward regime (iii), but its ability to scale to modern foundation models is still limited by memory and by the difficulty of capturing dense long-range dependencies.

## 3.2. Embedding and Latent Representation Design

Embedding compresses tokenized weights (and conditions) into latent variables that make generation tractable. This stage operationalizes two geometric facts from Section 2: (i) good solutions lie on a thin manifold, and (ii) the density on that manifold is highly non-uniform.

**Autoencoding and layer-wise balancing.** Early work showed that simple weight statistics predict performance (Unterthiner et al., 2020). Hyper-Representations (Schürholt et al., 2022) learn an autoencoder over entire networks, and emphasize *layer-wise loss normalization* so small layers are not ignored. SANE (Schürholt et al., 2024) produces layer-wise latents from large token sequences, enabling scalable encoding of large networks. Later, diffusion-style approaches often use an explicit encoder-decoder pair $E : \mathbb{R}^d \to \mathbb{R}^{d'}$ with $d' \ll d$ to run the generative model in latent space. (Wang et al., 2024) compresses a $\sim$100M ConvNeXt into a much smaller latent and performs diffusion in that space, mitigating the curse of dimensionality.

It has been noted that well-structured latents often admit semantic arithmetic: interpolation or vector arithmetic can produce predictable changes in behavior (Schürholt et al., 2022; Li et al., 2024b). It is a route to controllable generation and compositional model editing.

**Conditioning: from prompts and datasets to weights.** Conditioning is where "prompt-to-model" becomes concrete. Recent systems increasingly treat conditions as *task fingerprints*: a few prompts, a dataset descriptor, an image identity embedding, or an architecture graph.

D2NWG conditions diffusion on dataset descriptors to emit models for multiple target domains (Soro et al., 2025). Tina uses a CLIP text embedding of a user prompt to generate custom classifiers (Li et al., 2024b). HyperDream-Booth conditions on image embeddings to produce identity adapters (Ruiz et al., 2024). A notable recent step is prompt-conditioned *adapter generation* for LLMs: (Liang et al., 2025) learns a map from a small batch of unlabeled task prompts to LoRA updates, enabling task adaptation without per-task optimization. This strengthens the argument that conditioning need not be a handcrafted metadata vector; it can be directly extracted from raw task prompts.

## 3.3. Generative Predictors for Weight Synthesis

We now summarize the predictor families and take an opinionated stance on *where each is practically strongest*.

**Hypernetworks (fastest, best for personalization).** Classical HyperNetworks generate weights from low-dimensional codes (Ha et al., 2017). They remain the most pragmatic choice when latency matters and the output is parameter-efficient (e.g., LoRA-like updates). HyperDream-Booth exemplifies this: a lightweight hypernetwork gen-

erates low-rank personalization updates in seconds (Ruiz et al., 2024). In regime (i), hypernetworks currently offer the cleanest deployment path.

**Graph-conditioned predictors (cross-architecture, but scaling-limited).** GHN-style models predict weights from architecture graphs (Knyazev et al., 2021) and can generalize to unseen DAGs, enabling "zero-shot" evaluation in NAS-like workflows. Graph MetaNetworks (Lim et al., 2024) push toward broader architectural coverage. However, scaling these predictors to modern foundation models requires memory-safe graph representations and stronger inductive bias for long-range dependencies.

**Diffusion in weight space (expressive, currently best scaling evidence).** Diffusion-based generators perturb real weights with noise and learn to denoise. D2NWG (Soro et al., 2025) shows dataset-conditioned generation. Neural Network Diffusion demonstrates high-quality generation for large vision backbones in a compressed latent space (Wang et al., 2024). More recent hybrid diffusion stacks (e.g., recurrent encoders plus diffusion refinement) provide the strongest current evidence for mid-scale full-weight generation. We therefore view diffusion (particularly latent or hybrid diffusion) as the most promising direction for regime (ii), where distributional coverage and fidelity dominate.

**Normalizing flows (one-step sampling, promising but less mature).** Flow-matching approaches such as FLoWN offer invertible, one-step sampling of weights (Saragih et al., 2025), with potential advantages in controllability and likelihood-based evaluation. Their maturity and scale are currently behind diffusion, but they remain attractive where low-latency conditional sampling is critical.

**Hybrid stacks.** Hybrid predictors explicitly implement a curriculum: first model global inter-layer structure, then refine local details. RPG (Wang et al., 2025) is emblematic: a recurrent proto-encoder captures long-range dependencies, and diffusion fills token-level detail (Wang et al., 2025).

We note that RPG represents a milestone by efficiently synthesizing entire ConvNeXts or ViTs (up to 200M weights) in a single GPU pass - the first approach to go beyond toy 10M-parameter models to competitive 100M+ parameters, generated on commodity hardware. RPG-produced ConvNeXt-L matches conventional training within a few tenths of a percent on ImageNet. While other diffusion-based approaches (Wang et al., 2024; Soro et al., 2025) addressed scaling through modular generation, they often missed inter-block correlations, yielding suboptimal results.

### 3.4. Training Strategies and Collapse Avoidance

Training weight generators differs from standard generative modeling in two ways: (i) the data distribution is highly multi-modal (many tasks, architectures, and training recipes), and (ii) naive maximum-likelihood can collapse to memorization of a few basins. Practical training therefore relies on various stabilization techniques.

**Augmentation and regularization.** Basic noise injection and dropout over tokens mitigate memorization. MixUp-in-weight-space enlarges distributional support and improves generalization (Shamsian et al., 2024).

Contrastive objectives can encourage invariance to permutations and training artifacts (Schürholt et al., 2021). For INR generators, rendering-based fidelity losses (e.g., IoU penalties) provide functional supervision beyond raw weight error (Erkoç et al., 2023).

**Curriculum learning.** Practical curricula include shorter diffusion chains, smaller token sets, and progressively increasing context length, as in RPG (Wang et al., 2025). Conceptually, this corresponds to learning global structure first (easy) and token-level detail later (hard), matching the hierarchical dependence structure suggested by Section 2.

**Diverse model zoos and task coverage.** Diverse corpora reduce collapse by forcing the generator to explain multiple basins. D2NWG explicitly trains on heterogeneous checkpoint zoos (Soro et al., 2025). (Peebles et al., 2022) scale supervision dramatically, training on tens of millions of checkpoints to learn generative update rules. SANE aggregates multiple public model zoos and achieves strong zero-shot viability without overfitting (Schürholt et al., 2024).

### 3.5. Evaluation Metrics: Beyond One Simple Accuracy

A core critique of early weight-generation papers is that they over-index on task accuracy and under-specify how to evaluate generalization, novelty, and reliability. Evaluation must therefore be multi-axis, reflecting the fact that many distinct weights can yield similar accuracy but differ sharply in robustness, calibration, privacy risk, or mergeability.

**Three-Axis Framework.** We believe that any viable benchmark should report three axes as naturally induced by Fig. 1: (A) *within-architecture generation*, where we must test against nearest-neighbor retrieval, aligned interpolation, and model soups; (B) *conditional adaptation*, where prompt-, dataset-, or user-conditioned generators are compared against PEFT/fine-tuning under matched wall-clock and memory budgets; and (C) *cross-architecture generation*, where graph-conditioned generators are evaluated on held-out architecture families and unseen tensor schemas. Then, along each axis, evaluation should include downstream performance, generation cost, diversity and ensemble gain, calibration, robustness, memorization, provenance, and safety.

**Task performance and adaptation efficiency.** Accuracy (or BLEU / RL return) remains primary, but should be reported alongside adaptation cost: GPU-hours, wall-clock generation time, and memory footprint. For example, RPG-

generated ConvNeXt-L matches fully trained ImageNet accuracy in under 90 seconds (Wang et al., 2025). For personalization, *time-to-personalize* is often the key metric.

**Novelty vs. interpolation baselines.** A generator is considered novel only if it outperforms strong baselines such as aligned interpolation (Ainsworth et al., 2023) or model soups (Wortsman et al., 2022) at comparable distance. This concern is not hypothetical: recent diagnostic work reports that several representative weight generators produce replicas or simple interpolations of training checkpoints, and fail to outperform simple noise or ensemble baselines when novelty and performance are required jointly (Zeng et al., 2026). Representation similarity measures such as CKA (Kornblith et al., 2019) help detect trivial memorization, while spectral signatures (e.g., singular value distributions) test whether the generator matches the geometry of trained solutions.

**Diversity and mode coverage.** Since many solutions exist, generators should be evaluated for ensemble utility and mode coverage. Pairwise disagreement, ensemble gains, and performance variance across samples quantify whether the generator populates multiple basins.

**Robustness, calibration, and trust metrics.** Even within connected basins, models differ in robustness and calibration. Standard tools include NLL under corruption, ECE, and adversarial or stress-test evaluation. Model soups demonstrate that weight averaging can improve robustness without extra inference cost (Wortsman et al., 2022), motivating evaluation against such strong non-generative baselines. Depending on deployment, evaluation should extend to fairness and safety benchmarks such as trust and privacy assessment suites (Wang et al., 2023a; Li et al., 2024a).

## 4. Applications Enabled by Weight Generators

### 4.1. Instant Personalization

Weight generation has revolutionized model personalization across domains by amortizing adaptation into a single forward pass. In visual generation, HyperDreamBooth (Ruiz et al., 2024) dramatically accelerates subject-specific adaptation by producing weight updates in one pass, achieving personalization in ∼20 seconds (25× faster than Dream-Booth (Ruiz et al., 2023)) while preserving base-model knowledge and style diversity.

Beyond images, Tina (Li et al., 2024b) enables *text-to-model* generation, instantiating task-specific classifiers directly from textual descriptions using a single diffusion model. D2NWG (Soro et al., 2025) further generalizes this paradigm to dataset-conditioned full weight synthesis, matching standard transfer learning accuracy while improving few-shot performance by ∼6%, and extending to language models where it enhances LLaMA's mathematical reasoning. Most recently, (Liang et al., 2025) enable on-

the-fly task adaptation of LLMs: a user simply provides a description of a new task (e.g. a specific style of Q&A or a new programming API), and the system drops in a generated LoRA adapter that immediately makes the LLM perform well on that task. They demonstrated strong zero-shot generalization to tasks unseen during generator training, such as reasoning puzzles and multimodal queries, with performance even above that of standard fine-tuning.

### 4.2. Model Fusion and Editing

Generative weight models provide new tools for combining knowledge from multiple models or domains Traditional model fusion techniques typically require models to be trained on related data and carefully aligned in a shared coordinate frame, and even then produce only crude combinations that may underperform on complex tasks. In contrast, generative approaches can learn a weight-space distribution or *manifold* that encompasses diverse domains, enabling smooth interpolation or conditional sampling.

For example, (Peebles et al., 2022) showed that training a generative model on checkpoints from both image classification and language tasks yields latent interpolations that produce models with mixed characteristics, hinting at cross-domain fusion through learned manifolds. Recent work (Dravid et al., 2024) on interpreting the weight space of customized diffusion models demonstrates this phenomenon at scale: by populating a large dataset of fine-tuned diffusion checkpoints and projecting them into a low-dimensional subspace, they found that navigating within this learned subspace can generate new diffusion models encoding novel identities, and that linear directions in this space correspond to semantic edits (e.g., adding visual attributes) in the resulting models. This empirical evidence suggests that fine-tuned model weights behave as an interpretable meta-latent space where interpolation and controlled traversal produce meaningful, high-performing models.

### 4.3. Efficient Neural Architecture Search

Weight generation transforms Neural Architecture Search (NAS) by eliminating costly training of candidate architectures. By training hypernetworks that map architectures to performant weights (Brock et al., 2018), researchers can instantly predict weights for unseen architectures.

GHN (Knyazev et al., 2021), trained on one million neural nets, can predict on unseen and diverse networks, including all 24 million weights of a ResNet-50 in one forward pass, achieving 50% ImageNet top-5 accuracy and 60% CIFAR-10 top-1 without any SGD. These predicted weights serve as excellent initializations that can be quickly fine-tuned. Recent advances like GHN-3 (Knyazev et al., 2023) use Transformer-based hypernetworks to predict weights that outperform standard one-epoch SGD training, making "zero-shot" NAS feasible for exploring ultra-large design spaces.

### 4.4. On-Device Learning

Weight-generation methods also have important implications for on-device learning. By performing learning in a lightweight hypernetwork and deploying only its generated weights, the device does not undergo heavy training, and sensitive user data never leave the device. Several systems demonstrate its effectiveness. SecDOOD (Li et al., 2025) employs a cloud-based hypernetwork for video OOD detection on IoT devices, mapping encrypted feature summaries to specialized model weights without exposing raw data. In federated learning, pFedHN (Shamsian et al., 2021) utilizes a global hypernetwork to generate personalized models for each client, sharing only hypernetwork parameters while significantly outperforming traditional federated averaging and generalizing better to new clients.

## 5. The Next Frontier: Scaling Up Weight Generation to Foundation Models?

While synthesizing arbitrary trillion-parameter models end-to-end remains an open challenge, conditional weight generation has recently scaled far beyond toy settings and is advancing rapidly. It now operates on nontrivial large backbones, where deployment cost, adaptation latency, and repeated fine-tuning overhead are major practical bottlenecks.

The clearest evidence to date is **HY-WU** (Team et al., 2026), which recasts conditional weight generation as a *functional neural memory*: instead of repeatedly overwriting a shared parameter vector, a generator synthesizes instance-conditioned LoRA updates on the fly and injects them into a frozen backbone without test-time optimization. Notably, HY-WU is demonstrated on an 80B multimodal FM backbone with 13B active parameters, using an 8.11B generator to produce 0.72B LoRA parameters. This matters because it moves weight generation out of the "small-model curiosity" regime and into a setting where conditional weight generation is already interfacing with genuinely large-scale foundation-model infrastructure in a leading industry lab.

This evidence is reinforced by a broader scaling trend. RPG (Wang et al., 2025) generates competitive ConvNeXt and ViT backbones with up to $\sim$200M parameters by explicitly modeling inter-layer dependencies. Text-to-LoRA (Charakorn et al., 2025) generates LoRA updates for Llama-3.1-8B and Gemma-2-2B; Doc-to-LoRA (Charakorn et al., 2026) does so for Gemma-2-2B, Mistral-7B, and Qwen3-4B; and SHINE (Liu et al., 2026) extends the regime to Qwen3-8B. These results suggest a consistent staged picture: today's strongest evidence is not unrestricted full-checkpoint generation at frontier scale, but large-backbone, adapter-scale, and conditional weight generation.

At the same time, the **main barriers** to scaling are not merely matters of raw compute, but of structure. Represen-

tation alignment, long-range dependencies, and checkpoint heterogeneity are fundamental technical challenges rather than secondary engineering details. These factors will likely determine whether the current adapter-scale successes can eventually extend to broader full/base model-generation regimes. We discuss these issues below:

**i. Representation alignment & symmetry.** Raw weights are not canonical. Permutation and scaling symmetries create many numerically distinct parameterizations of the same function. Without alignment, quotient-aware encoders, or symmetry, a generator may waste capacity modeling representation artifacts rather than functional variation.

**ii. Long-range dependency and memory scaling.** Large models contain cross-layer and cross-module dependencies that are poorly captured by naive flattening or purely local chunking. Scaling likely requires (more) hierarchical tokenization, latent or recurrent generators, sparse graph attention, and scoped generation of parameter subsets rather than monolithic modeling of all tensors at once.

**iii. Architecture and training heterogeneity.** Checkpoints differ by architecture, tokenizer, preprocessing, optimizer, scheduler, regularization, objective, data mixture, and fine-tuning recipe. Treating all checkpoints as samples from one unconditioned density is unlikely to succeed. A more realistic target is the conditional family $p(W \mid A, C, R)$ (Fig. 1). In practice, such metadata should be used for filtering, stratification, or conditioning, rather than ignored.

**iv. Memorization, provenance, and safety.** A generator trained on checkpoint zoos can inherit proprietary weights, backdoors, biases, or contamination. Weight generation will therefore face a "model-supply-chain" problem: provenance, lineage tracking, memorization tests, safety audits, and quarantine of untrusted checkpoints.

## 6. Critical Discussions

We clarify what weight-space generation *can* and *cannot* currently do, and confront credible counter-positions.

### 6.1. Can Generation Go Beyond Optimizer Solutions?

A central question is whether weight generators merely reproduce the outcomes of SGD or can synthesize *novel* solutions. At present, most empirical results show parity with well-tuned optimization rather than consistent superiority. However, novelty should be evaluated structurally rather than numerically. Generative models are trained across many checkpoints and tasks, and thus have access to cross-model correlations unavailable to single-run optimization. Evidence from generative checkpoint models (Peebles et al., 2022) and from interpretable diffusion subspaces (Dravid et al., 2024) shows that generators can sample distinct regions within the same connected basin, differing in robustness, calibration, or semantic attributes. Whether such diver-

sity can be systematically exploited to outperform SGD on core metrics remains an open question, but the mechanism, namely sampling from a learned density rather than following a single trajectory, is fundamentally different. Recent analyses caution that current generators do not automatically achieve such distributional novelty (Zeng et al., 2026).

## 6.2. The Limits of Weight Space Learnability

Another concern is whether weight space exhibits sufficient locality or regularity to be learnable at scale. We make clear that locality in weight space is *architectural*, not spatial. Dependencies are long-range, module-based, and symmetry-entangled. This is why naive token-wise models fail and why structured tokenizers, hierarchical generators, and curriculum learning are necessary. Modularity offers promise for compositional generalization, but boundaries between modules are neither smooth nor universal. Hence, generation is feasible but non-trivial: it requires inductive biases that mirror those of successful optimization. Without sufficient checkpoint diversity, and held-out tests against retrieval, interpolation, or perturbation baselines, a generator may learn a compressed description of its training zoo rather than a reusable model distribution (Zeng et al., 2026).

## 7. Alternative Views

▷ **Alternative View 1:** *"Fine-tuning, PEFT, Transfer Learning, and In-Context Learning are Sufficient."* One might argue that PEFT, LoRA, transfer learning, and in-context learning already provide fast and practical adaptation. We agree that these paradigms remain attractive when iterative per-task optimization is acceptable, task data are available, and strong task-specific adaptation is desired. In-context learning is preferable when no persistent parameter change is needed. Our response is therefore economic rather than purely algorithmic: weight generation targets settings where the cost of repeated optimization or long context windows is itself the bottleneck, for example in low-latency or privacy-constrained deployment. Its goal is to amortize that cost across tasks and users, enabling second-scale adaptation when deployment-time optimization is undesirable.

Continual learning is another adjacent but distinct paradigm. It focuses on updating one deployed model over time while avoiding forgetting. By contrast, weight generation aims to learn a distribution over reusable parameter updates or checkpoints that can be sampled without committing all knowledge to a single evolving parameter vector. It does not solve forgetting by itself, but it could complement continual learning by generating task- or episode-specific updates while keeping provenance and rollback explicit.

▷ **Alternative View 2:** *"Yet Another Meta-Learning or Model Merging Under a New Name."* Weight generation overlaps with both meta-learning and model merging, but the primitive it studies is different. Meta-learning typically operates over tasks with shared structure and still culminates in iterative adaptation. Model merging is closer in spirit, but usually assumes a small set of compatible source models and simple algebraic composition. By contrast, weight generation treats *trained models themselves* as data and asks whether useful parameter updates can be produced in a single feed-forward pass from conditioning information. This distinction matters for evaluation, infrastructure, and governance. We do not claim that weight-space generation already dominates these approaches, but rather that it opens a distinct amortized one-shot regime that remains under-standardized relative to its promise.

▷ **Alternative View 3:** *"Weight Generation Will Not Matter Until It Outperforms Gradient-Based Optimization End-to-End for Full Foundation-Model Training."* We do not believe this is the right threshold. A new primitive can become foundational without immediately replacing the entire optimization stack. It may first prove its value by replacing or amortizing expensive subroutines that recur throughout modern ML pipelines. In our view, the strongest current case for weight generation lies precisely in this regime: adaptation, personalization, context internalization, and model-space search, where current practice repeatedly invokes optimization to produce relatively small but operationally valuable weight updates. From this perspective, frontier-scale full-model synthesis remains an important open research problem rather than the central bar that the field must clear immediately. At the same time, as discussed in §4.5, there is growing evidence that the field is making meaningful progress toward larger-scale regimes.

## 8. Conclusion and Call to Action

This paper argues that neural network checkpoints should be treated not only as optimization outputs, but as a first-class data modality for generative modeling. We believe the path forward is not a single universal weight generator, but a standardized research stack. *First*, the community needs checkpoint-as-data infrastructure: curated model zoos with architecture schemas, training recipes, quality traces, provenance, licensing, and dataset lineage. *Second*, WSG methods must respect the structure of weight space, including permutation symmetry, low-rank and modular structure, optimizer-induced bias, and long-range cross-layer dependencies. *Third*, evaluation must move beyond task accuracy to multi-axis benchmarks that measure efficiency, novelty beyond retrieval or interpolation, diversity, robustness, calibration, memorization, and safety. *Fourth*, public weight generators should include provenance tracking, watermarking, privacy audits, and safety screening as part of the release protocol rather than as afterthoughts. If these standards are built, weight-space generation can turn repeated artisanal optimization into a reusable and auditable model-production process. The field should advance through deliberate infrastructure, benchmark, and governance design.

## Impact Statement

This paper advocates treating neural network checkpoints as a first-class data modality and standardizing generative modeling in weight space as a core machine learning primitive. If adopted, this paradigm could substantially reduce the computational and environmental cost of repeated model training, enable rapid and privacy-preserving personalization, and lower barriers for deploying specialized models on edge and resource-constrained devices.

At the same time, generative access to model weights raises legitimate concerns around memorization, misuse, and model provenance. A generator may memorize proprietary weights, raising IP and copyright concerns analogous to data leakage in LLMs (Li et al., 2024a). Automated backdoor insertion or mass production of harmful models are also plausible misuse scenarios. Mitigations will require provenance tracking, watermarking of generated weights, controlled access to training corpora, and evaluation protocols that test for memorization and malicious behavior. These concerns strengthen the case for rigorous community standards. Overall, the anticipated societal impact is dual-use: weight-space generation has the potential to democratize and accelerate AI deployment, but only if accompanied by appropriate governance and safeguards.

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
