# OpenReview forum: "Position: Weight Space Should Be a First-Class Generative AI Modality"
_ICML.cc/2026/Position_Paper_Track — ICML 2026 Position Paper Track regular_

### Official Review · Reviewer_prJ2 · 2026-03-09

**Significance:** 3
**Argument Clarity:** 2
**Rating:** 4
**Confidence:** 4

**Questions:**

1. What are the primary technical bottlenecks that prevent scaling the proposed approach to large heterogeneous models and how do the authors envision to overcome them?

2. How robust is the proposed paradigm to heterogeneity of training pipelines, such as different optimizers, data preprocessing pipelines, regularization methods, training objectives across checkpoints, etc.?

**Alternative Views Section:**

Yes

**Compliance With Llm Reviewing Policy A Conservative:**

Affirmed.

**Discussion Potential:**

4

**Final Justification:**

I'm positive about the paper, provided that the clarifications made in the rebuttal are incorporated into the final version.

**Paper Summary:**

This paper argues that neural network checkpoints should be treated as an important data modality and that generative modeling in weight space should become a core machine learning primitive. The paper summarizes theoretical insights on weight-space geometry, organizes existing approaches into a pipeline, surveys recent applications relevant to the proposed position, and proposes a research agenda to support this paradigm.

**Position:**

Yes

**Position In Title:**

Yes

**Related Work:**

4

**Strengths And Weaknesses:**

**Strengths**
* A clear position advocating for generative modeling in weight space as a foundational paradigm for future machine learning systems.
* A timely and relevant topic, given the growing number of publicly available checkpoints and recent work on model editing, hypernetworks, diffusion-based weight generation.
* The manuscript is well-organized and readable, and the conceptual framing of model creation  by sampling from a distribution of weights instead of repeatedly optimizing is thought-provoking.
* The discussion of motivations in Section 2 is a useful synthesis of literature on weight-space geometry, including mode connectivity, permutation symmetries, flatness, and implicit bias. The attempt to structure the field through a standardized pipeline is valuable as a conceptual framework.
* The paper also identifies several application areas, such as personalization, model fusion, architecture search and on-device, where weight-space generation could plausibly reduce training cost and latency.
* The topic is relevant to the ICML community and could potentially inspire discussion.

**Weaknesses**
* The paper currently reads more as a survey combined with an advocacy piece than as a position paper that critically analyzes the challenges of the proposed paradigm. The argumentation supporting the central claim is somewhat one-sided. While the manuscript lists theoretical motivations and recent empirical demonstrations, it does not provide in-depth analysis of the fundamental obstacles that may limit the practicality of weight-space generation. For example, challenges such as distribution shift across checkpoints, incompatibility between training recipes, optimizer dynamics, architectural heterogeneity are only briefly mentioned or not discussed in depth. A position paper would benefit from a more explicit analysis of the conditions under which the proposed paradigm may fail or become impractical.
* The discussion of alternative viewpoints is relatively shallow. The paper mentions several counterarguments, but these sections are short and do not engage deeply with opposing perspectives. In particular, it would be useful to more carefully compare weight-space generation with existing paradigms such as meta-learning, transfer learning, PEFT, and model merging, including situations where those approaches may remain preferable.
* The manuscript does not clearly identify the most important unsolved technical problems to make the proposed position work. For example, it would be valuable to discuss open challenges such as heterogeneous architectures, avoiding memorization of proprietary models, bias, heterogeneity of training pipelines, etc.
* Some of the empirical evidence cited appears preliminary and the paper occasionally extrapolates broad conclusions from limited demonstrations. While the authors acknowledge this partially, the paper could more carefully distinguish between demonstrated capabilities and speculative future potential.

**Support:**

3

---

> ### Author Rebuttal · Authors · 2026-03-29
>
> Thank you and we agree with the core of your comments. We do not intend to claim that weight-space generation already replaces optimization in general. Rather, it is that checkpoints have become a meaningful data modality, and the next step is to standardize the field around the conditions under which WSG works, fails, and can be responsibly evaluated. That is why we see this as a position paper rather than a survey: the central claim is normative, namely that the community should organize benchmarks, metadata, and infrastructure around the blockers, not treat current positive results as isolated anecdotes.
>
> Concretely, we will expand the current discussion into a clearer bottleneck section organized around four blockers:
> - *Representation alignment / symmetry*. Raw weights are not canonical because of permutation and scaling symmetries. Without symmetry-aware encoders or canonicalization, generators waste capacity modeling equivalent parameterizations
> - *Long-range dependency &memory scaling*. Large models exhibit strong cross-layer and cross-module dependencies, so naive flattening or local chunking scales poorly. This is why adapter generation is more mature than full end-to-end generation, and why hierarchical / recurrent / latent approaches appear necessary
> - *Architecture & training pipeline heterogeneity*. Ckpts differ not only by task, but by architecture graph, optimizer, regularization, augmentation, preprocessing/tokenization, and fine-tuning recipe. Treating all such models as samples from one unstructured density is unlikely to work well. A more realistic target is a conditional family p(W | A, C, R), where A is architecture, C is task/domain/user condition, and R is recipe/quality metadata. Importantly, R is mainly a corpus-side variable used to stratify, filter, or condition training/evaluation; it need not be a user-facing input at deployment. A user may specify only a high-level condition, while the generator marginalizes or internally selects suitable recipe factors
> - *Memorization, provenance & safety*. A generator trained on checkpoints can inherit proprietary information, backdoors, or biased behavior from the source zoo. This is a real technical obstacle, and we will elevate it from the ethics discussion into the main agenda
>
> **To your Q1 (bottlenecks & path forward)**: our answer is essentially the four blockers above. The practical path forward is not a monolithic universal generator but a staged strategy: symmetry-aware representations, architecture-conditioned encoders (especially graphs when A varies), hierarchical generation for long-range dependencies, metadata-conditioned training to handle heterogeneity, and explicit provenance/safety auditing
>
> **To Q2 (robustness to training heterogeneity)**: we agree robustness does not come for free. Different optimizers, preprocessing choices, objectives, and regularizers induce a more multi-modal ckpt distribution. We do not think this heterogeneity is fatal, but it does mean the field needs better metadata and conditioning. We will revise to clarify: when heterogeneity is modest, a shared latent may work; when it is large, recipe metadata, stratified corpora, or MoE style generators become necessary
>
> We also agree **the alternative-view should be deeper**. In revision, we will expand the comparison to PEFT, ICL, meta-learning, transfer learning, and model merging. In particular, PEFT / transfer / most meta-learning pipelines remain preferable when iterative per-task optimization is acceptable, data are available, and stronger task-specific adaptation is desired; ICL remains preferable when no persistent parameter change is needed; and model merging remains attractive when suitable source models already exist and simple composition suffices. Our targeted distinction is that these paradigms generally do not study the same primitive as WSG: except for model merging, they typically still culminate in iterative optimization, whereas WSG asks whether useful parameter updates can be produced in a single feed-forward pass. Model merging is closer in spirit, but usually depends on a few compatible source models and does not by itself define a general conditional generator over weight space. We don't claim WSG already dominates these approaches, but that it opens a distinct amortized one-shot regime that is currently under-standardized relative to its promise.
>
> Finally, we will more **carefully map the demonstrated capabilities**, stratified by model size and granularity. Adapter generation, personalization, and conditional weight synthesis on large backbones will be described as practical today, while foundation-scale end-to-end full-ckpt synthesis remains open. A few newer results (after ICML ddl) make this boundary sharper than before: HY-WU reports an 8B generator attached to an 80B backbone with 13B active parameters, generating up to 0.72B LoRA parameters on the fly; Doc-to-LoRA on Gemma-2-2B, Mistral-7B, and Qwen3-4B; and SHINE on Qwen3-8B.

---

> > ### Author Rebuttal · Reviewer_prJ2 · 2026-04-01
> >
> > I've decided to increase my score, provided that the suggestions listed in the rebuttal are incorporated.

---

### Official Review · Reviewer_j5ih · 2026-03-11

**Significance:** 3
**Argument Clarity:** 4
**Rating:** 5
**Confidence:** 3

**Questions:**

* What mechanisms can ensure that WSG doesn't propagate biases from the checkpoints used to train the generator? How can we ensure that we are leveraging "safe" checkpoints when training the generator?
* Do we need to advocate for open-data along with better checkpoint metadata? Should this be explicitly part of the call to action?
* What specific model checkpoint metadata would be required to improve the viability of WSG?
* Are there any preliminary research directions regarding overcoming the memory constraints of graph-based tokenization?

**Alternative Views Section:**

Yes

**Compliance With Llm Reviewing Policy A Conservative:**

Affirmed.

**Discussion Potential:**

4

**Final Justification:**

The suggested revisions will strengthen the paper and further support the author's position. I have maintained my inital rating of Accept.

**Paper Summary:**

This position paper argues that the time has come to elevate model checkpoints to a first-class data modality and that weight space generation (WSG) should become a standard strategy for producing specialized or task-specific models rather than traditional optimization techniques. A detailed literature review of recent findings regarding the geometric properties of the loss landscape and parameter space support the authors’ argument that the high-quality solutions lie on structured, low-dimensional manifolds which make parameter generation tractable. The paper includes a summary of current practical approaches in WSG including tokenization methods, embeddings, and generator architectures. A nuanced discussion regarding where WSG is currently working well (LoRA adaptation) versus where open research questions remain (WSG at billion parameter scale, OOD DAGs)  is also included. The paper concludes with reasonably strong counter arguments.

**Position:**

Yes

**Position In Title:**

Yes

**Related Work:**

4

**Strengths And Weaknesses:**

## Strengths
* Detailed literature review of both current WSG approaches and the supporting theory that argues in favour of high quality solutions occupying a low-dim. manifold in weight space.
* Well written, the paper is a compelling read. Summarizing sentences and paragraphs are used to excellent effect.
* The 5-stage pipeline of WSG and review of existing approaches to each step of the pipeline helps establish current practical approaches.
* The paper is timely, only in recent years have model checkpoints been made publicly available at scale on platforms such as HF and modelscope.
* Avoids overstating claims, acknowledges many approaches still lag significantly behind standard iterative gradient based optimization.

## Weaknesses:
* One of the fundamental barriers to leveraging model checkpoints at scale is the lack of associated metadata. While the authors suggest that standardized metadata should be included with the publication of model checkpoints, they do not offer any concrete recommendations here as to what metadata is crucial to ensuring the utility of checkpoints in the WSG paradigm.
* Further to the above, dataset metadata is also lacking standardization. It may be that before we can leverage model checkpoints at scale we require more model checkpoints with truly open data.
* WSG may introduce a new attack vector for model poisoning. The typical approach of only releasing open weights means it will likely be even more challenging to isolate or remove training set contamination, inherent biases, or other unwanted behaviour when generating models with WSG. Recent work [1]  has found that using KL loss from a biased teacher can impart that bias on the student, even without explicitly including contaminated data during the KL process. It's unclear if WSG also propogates bias in this manner.
* The authors highlight that evaluation metrics should go beyond just accuracy. However, in terms of predicting ML trends it has been the historic case that it is only when a method yields superior generation that it is broadly adopted. As such, it may be that WSG will emerge as a foundational operation in ML if and only if it offers superior accuracy to current gradient based optimization approaches. While such potential appears to exist in predicting small weight updates for adaptation, it’s unclear if WSG will be able to scale to foundation model sizes or beyond model adaptation.
* A more detailed summary of the challenges of WSG at scale would help inform the specifics of algorithmic or memory limits that inhibit this scaling.
* A concrete proposal of a benchmark suite that meets the goals of section 3.5 would help standardize the WSG field.

## Typos
* L198: "optimizatio"

[1]  H. Chaudhari, J. Hayes, M. Jagielski, I. Shumailov, M. Nasr, and A. Oprea, “Cascading Adversarial Bias from Injection to Distillation in Language Models,” Oct. 05, 2025, arXiv: arXiv:2505.24842. doi: 10.48550/arXiv.2505.24842.

**Support:**

3

---

> ### Author Rebuttal · Authors · 2026-03-29
>
> We thank the reviewer for the thoughtful and constructive feedback. We agree that Sec. 3.5 should become more concrete, and in revision we will strengthen it along five axes.
>
> **Metadata and open-data**. We agree that checkpoint utility depends critically on metadata quality. In revision, we will add a concrete checklist for checkpoint release, including: architecture graph / tensor schema, tokenizer, data provenance and licensing, data-mixture summary, training recipe and major HPs, optimizer / scheduler / batch size / regularization, evaluation traces, and lineage linking a checkpoint to its parent models and data sources. We also agree that dataset metadata matters. We will make our call explicit: not “open weights alone are enough”, but that WSG will benefit most from joint progress in checkpoint metadata, dataset documentation, and, where possible, more open-data training pipelines.
>
> **Safety, bias propagation, and poisoning**. We agree this deserves a sharper discussion. WSG introduces a model-supply-chain risk: if the source zoo contains biased, poisoned, or contaminated checkpoints, a generator may inherit or even recombine those failure modes. We will make this explicit, and revise the paper to advocate not only broader checkpoint access, but also safe checkpoint curation, including provenance tracking, data / recipe lineage, safety evaluations, and exclusion or quarantine of untrusted checkpoints.
>
> **Does WSG matter only if it beats gradient optimization end-to-end?** We think this is the central practical question. Our position is not that WSG must already replace gradient-based optimization for full foundation-model training in order to become foundational. A new primitive can become foundational by first replacing or amortizing repeated expensive subroutines of the ML stack. We believe the strongest current case for WSG is exactly here: adaptation, personalization, context internalization, and model-space search, where today we repeatedly run optimization to produce relatively small but valuable weight updates.
>
> Importantly, this regime is no longer limited to toy models. Recent work already demonstrates conditional weight generation on nontrivial backbones, including HY-WU (released in March 2026 after ICML ddl, the most scaled to date) on an 80B multimodal FM backbone (13B active parameters) with an 8.11B generator producing 0.72B LoRA parameters; Text-to-LoRA on Llama-3.1-8B and Gemma-2-2B; Doc-to-LoRA on Gemma-2-2B, Mistral-7B, and Qwen3-4B; and SHINE on Qwen3-8B. We will revise the paper to make this staged view explicit: today’s strongest evidence is for adapter-scale / conditional weight generation on large backbones (we will cite all these newer/post-ICML ddl evidences), while unrestricted full-model generation at frontier scale remains an open research problem rather than our central claim.
>
> **Challenges and benchmarks at scale**. We agree that the paper should separate the main scientific bottlenecks from the evaluation agenda more clearly. In revision, we will summarize the main blockers more explicitly: canonicalization / symmetry in weight space, architectural heterogeneity, missing metadata, long-range dependency and memory cost in tokenization, and safety / memorization risks. We will also add a concrete benchmark sketch with three axes: (A) within-architecture generation, (B) conditional adaptation from prompts, datasets, or user conditions, and (C) cross-architecture generation for unseen architecture graphs. For each axis, evaluation should include not only downstream performance, but also efficiency, novelty beyond retrieval / interpolation / model soups, diversity and ensemble gain, robustness / calibration, and memorization / safety.
>
> **Graph-tokenization memory limits**. We agree this deserves more explicit treatment. Promising directions include more structured tokenization, hierarchical or blockwise generation, sparse / local attention over architecture graphs, and scoped generation of parameter subsets or low-rank updates instead of full dense weights. We will make these directions explicit in the revised discussion.
>
> We appreciate the typo catch and will fix it.

---

> > ### Author Rebuttal · Reviewer_j5ih · 2026-04-03
> >
> > The suggested revisions will strengthen the paper and further support the author's position.

---

### Official Review · Reviewer_MMf1 · 2026-03-12

**Significance:** 3
**Argument Clarity:** 3
**Rating:** 4
**Confidence:** 4

**Questions:**

1. How can different network architectures be unified within a single framework?
2. Should models with the same architecture and task but different performance levels be trained together?
3. Why is Graph tokenization the clearest path toward regime (iii)

**Alternative Views Section:**

Yes

**Compliance With Llm Reviewing Policy A Conservative:**

Affirmed.

**Discussion Potential:**

3

**Final Justification:**

My main concern with this paper lies in its presentation. The authors’ clarification in the rebuttal effectively addressed my confusion and provided a detailed plan for improving the presentation. This has led me to update my assessment to be more favorable toward acceptance.

**Paper Summary:**

The paper proposes treating model weights themselves as the target of generation, and provides the background, clear motivation, methodology, and a potential roadmap supporting this perspective. The authors also discuss the feasible and infeasible settings under the current paradigm, helping clarify the practical scope and limitations of this direction.

**Position:**

Yes

**Position In Title:**

Yes

**Related Work:**

2

**Strengths And Weaknesses:**

### Strength

1. The paper provides a clear and well-articulated background, and the motivation for the proposed perspective is easy to understand. This helps readers quickly grasp the problem setting and the rationale behind treating model weights as a generation target.
2. The paper presents its arguments in a clear and logically structured manner. The claims are supported by theoretical reasoning, making the overall argumentation convincing.
3. The paper presents a unified framework that organizes the current progress in this area while also highlighting key unresolved problems. This structure helps clarify what has already been achieved and what remains to be addressed in future research.

### Weakness

1. The paper takes a relatively long time to introduce the main point. After presenting the background and motivation, it would be more effective to clearly state the paper’s central **position** before proceeding with the detailed analysis. This would help readers quickly understand the main claim and better follow the subsequent discussion.
2. The current presentation remains relatively high-level and may be difficult for readers to grasp intuitively. Providing concrete examples and illustrative figures would help clarify the concepts and make the discussion more accessible.
3. The presentation of the sections feels somewhat fragmented, even though there appear to be strong dependencies among them. For instance, **graph-based methods** may be capable of generating parameters for different architectures, whereas **diffusion-based methods** seem more naturally suited to generating different parameter instances within the same architecture (otherwise it is unclear how the generated parameters would be mapped back to model weights). Clarifying these relationships would help readers better understand the overall framework. Additionally, the descriptions of the underlying mechanisms remain quite abstract. Including more **illustrative figures and concrete examples** would make the methodology easier to follow.

**Support:**

3

---

> ### Author Rebuttal · Authors · 2026-03-29
>
> Thank you for the thoughtful review. Your main point is well taken: the core position can be presented much more clearly by stating it earlier and by making the regimes and their relationships more concrete.
>
> **Presentation and structure**. We will move the central position statement to the end of the first introduction paragraph, before the longer motivation, and add both a simple “map of the field” figure and a running example. The figure will separate three regimes that are already present in the text but not visualized clearly enough: (1) fixed architecture, varying weights; (2) fixed architecture, varying task/domain/user conditions; and (3) varying architecture plus varying weights. We will then anchor representative methods to these regimes, e.g. hyper-representation / p-diff / RPG-type methods in (1), prompt- or dataset-conditioned update/weight generation in (2), and GHN / GMN-style methods in (3).
>
> **Relationships among method families**. These are not disconnected categories, but different components of a common factorization. The key distinction is between methods that define the structure and placement of parameters, and methods that generate the values of those parameters. When architecture is fixed, the parameter slots are already known, so diffusion- or hypernetwork-style generators can operate directly over weight coordinates or low-rank updates. When architecture varies, the first challenge is to encode the computation graph and determine how generated parameters map back to operator slots, tensor shapes, and connectivity. This is why graph-based methods become natural in regime (3). Put simply, graph methods answer “where do the weights go?”, while diffusion / hypernetwork methods answer “what values should they take?” In practice, the most natural systems are often hybrid: a graph-based encoder to represent architecture, followed by a continuous generator for the parameters conditioned on that encoding.
>
> **How can different architectures be unified within a single framework?** Our intended unifying view is a conditional family p(W|A,C,R), where W are weights, A is an architecture description, C is task/domain/user condition, and R is quality/recipe metadata. This also clarifies the three regimes. When A is fixed and C is fixed or implicit, the task is to model variation in W for a single template; this mainly tests representation of a structured weight distribution (e.g., p-diff-style). When A is fixed but C varies, the generator performs conditional adaptation over a known parameter layout, as in prompt-, dataset-, or user-conditioned updates (e.g., DnD, Doc2LoRA, HY-WU).
>
> When A varies as well, the challenge is no longer only what values to generate, but where they should go. A must be encoded explicitly, e.g., as a computation graph whose nodes and edges capture operator types, tensor shapes, and connectivity. This is why graph-based methods become especially important in regime 3, while diffusion/ hypernetwork modules remain natural choices for generating parameter values conditioned on that architectural encoding, leading to a hybrid template “architecture encoder + condition/metadata encoder + weight generator.” Despite existing work in this regime (e.g., GHN-style ), their performance still does not yet match networks obtained by standard training, highlighting open challenges.
>
> **Should models with the same architecture and task but different performance levels be trained together?** Yes, but only with curation or explicit quality metadata. Mixing checkpoints of different quality can be useful because it exposes the geometry of training, but if the target is high-performing generation, quality should either be filtered for or exposed as conditioning, for example validation accuracy, training stage, or recipe. Otherwise the generator may average across incompatible regions and blur the target density. We will make clearer that the paper is advocating curated, quality-annotated checkpoint corpora rather than arbitrary dumps of weights.
>
> **Why is graph tokenization the clearest path toward regime (3)?** Because once architecture varies, the challenge is no longer only the numeric value of parameters, but the relational structure telling us where those parameters belong. Graphs are the most direct representation of this: they encode operator identity, tensor shapes, and connectivity, and they respect architectural symmetries better than flat sequences. That said, we do not mean graph methods only. A likely practical recipe is hybrid: use a graph-based encoder for A, then diffusion- or hypernetwork-style generation for W conditioned on that encoding. We will make this relationship explicit and add concrete running examples showing how generated parameters are mapped back to model weights in both single-architecture and cross-architecture settings.
>
> Finally, thank you for the suggestion on figures and examples. We agree they would materially improve accessibility, and we will add them.

---

> > ### Author Rebuttal · Reviewer_MMf1 · 2026-04-03
> >
> > Thanks for the authors’ clarification. I will accordingly raise my score. I now have a much clearer understanding of the paper’s main focus and overall approach. I encourage the authors to further refine and polish the writing to improve readability.
> >
> > Finally, it would be valuable if the authors could briefly discuss the differences and connections between Weight Space Generation and continual learning, and whether the Weight Space Generation paradigm could be extended to address the core challenge of continual learning in AI.

---

### Official Review · Reviewer_FGfh · 2026-03-16

**Significance:** 4
**Argument Clarity:** 4
**Rating:** 5
**Confidence:** 3

**Questions:**

Please see Q1-3 above.

**Alternative Views Section:**

Yes

**Compliance With Llm Reviewing Policy A Conservative:**

Affirmed.

**Discussion Potential:**

3

**Final Justification:**

The questions raised in this review were fully resolved and other discussions also appear to be resolved and lead to planned improvements. It reinforces my prior assessment and I recommend accepting.

**Paper Summary:**

The paper systematically advocates that the space of the weights of trained models is interconnected across architectures and tasks and that there are many potential benefits in modelling this manifold with machine learning tools, for instance making feasible a zero-shot model adaptation to a specialized task or private setting (distribution shift) via direct conditional generation of model weights instead of fine-tuning, similar to meta-learning. The main position of the paper is to promote systematic collection of trained weights as data points, and promote modelling these weight manifolds with generative models. The paper outlines the theoretical reasoning, discusses current practical mechanisms, while organizing them in a  taxonomy and relating with the theoretical reasoning and discusses potential applications.

**Position:**

Yes

**Position In Title:**

Yes

**Related Work:**

4

**Strengths And Weaknesses:**

The paper is exceptionally well written and articulated. The paper makes a systematic argument, based on the well-organized and concise analysis of large body of works studying solution manifolds, inductive biases, cross-network combinations, recent advances on weight generation, current and potential applications, that the problem is feasible, has a potential to unlock much more efficient solutions for a range of problems, yet challenging. It emphasizes the importance of standardized data infrastructure, developing generative models that would be efficient w.r.t. symmetries in the model spac, thorough evaluation, and responsible deployment. I think it is of an appropriate contents and credibility for a position paper.

It seems somewhat unintuitive and perhaps unnatural that the weight space is first discussed without the regard for the respective architecture. Later on this is refined, and the problems are split into per-architecture generation and those that give it a more general meaning by performing architecture encoding and weight positioning in it in some form.

**Q1** In the discussion of the connectedness of the solutions, does the mentioned studies apply to classification problems (where one may expect zero training error in the overparameterized mode) or regression problems as well? What is referred to as solutions in this context: the models attaining zero / low training loss or the ones that attain good test performance? I.e. does the paper mean to say that well-generalizing solutions form a manifold?

**Q2** In the discussion of flatness, the paper claims that "most directions of trained models exhibit near-zero curvature" (implies perturbing them retains the performance) and that the "effective dimension of the solution space is far smaller than the ambient parameter count" -- isn't there a contradiction? Perhaps you mean co-dimension of the solution manifold?

**Q3** I found it somewhat unclear what the generative task in some settings actually is. In particular the paper highlights that RPG (Wang et al. 2025) can generate high-accuracy ImageNet1k models. However, it is trained on high accuracy ImageNet 1k models. It generates 200M- parameter ConvNeXt-L models using a 200M-parameter recurrent diffusion model. It is not apparent that it learns the distribution of all good models. The paper mentions measuring ensemble advantage. Even considering that, a generative model could be a simple mixture of per-basin means (suitably compressed) with some structured noise directions, and achieve a high accuracy and likely show an ensemble advantage. So what is actually the purpose of learning a generative model for a fixed architecture on a fixed dataset? That would also define the suitable evaluation criterion. Does it reduce / not reduce to just compression of the models in the training set? Does it make sense without some kind of generalization over architectures or the data distribution?

2.1 "trained neural networks do not occupy isolated optima" -- is it really despite extreme overparameterization or because of?

### Some typos:
lines 185, 198, 299

**Support:**

4

---

> ### Author Rebuttal · Authors · 2026-03-29
>
> Thank you for the careful and very constructive review, and for the positive assessment. We are glad the paper’s core position and organization came across as credible. Your comments help us tighten several statements that are currently broader than they should be.
>
> Q1. You are right that our wording on connectivity was too broad. In the cited literature, the safest statement is about low-training-loss / low-objective SGD solutions, or more generally low-loss sublevel sets, with much of the empirical evidence coming from classification. Some papers also show that test performance stays nearly constant along connecting paths, but that still does not justify the stronger claim that “all well-generalizing solutions form a manifold.” In revision we will make this precise: connectivity is evidence that good solutions are often not isolated, while generalization remains an empirical property of some connected regions rather than a theorem about the entire solution set.
>
> Q2. We agree the current phrasing risks conflating two different notions. “Flatness” is local: around a trained point, many nearby directions may have small curvature. “Low effective / intrinsic dimension” in the subspace-training literature is about the number of effective degrees of freedom needed to reach strong solutions during optimization. These are compatible. A model can lie in a broad low-loss region while the optimization trajectories or effective search directions concentrate in a much smaller subset than the ambient parameter space. We will revise the text to say “intrinsic dimension of the optimization process” rather than “solution space” at that point, and to separate local curvature, effective optimization dimension, and codimension, without over-interpreting Hessian evidence as a global statement about the manifold of all good solutions.
>
> Q3. We agree this distinction is important, especially for fixed-architecture / fixed-dataset settings such as RPG. Our target claim is not that such a model has already learned “the distribution of all good models.” Rather, this regime is a controlled testbed for learning a nontrivial slice of the solution distribution induced by realistic training pipelines. Its value is not only compression: (i) one-shot synthesis can replace repeated optimization, and may also provide a principled way to amortize across heterogeneous optimization HPs or training recipes, including runs produced by different optimizers such as AdamW and Muon; in our recent preliminary observations, such heterogeneity can enrich the diversity of the learned checkpoint family and lead to generating more "novel" solutions, (ii) it supports diverse near-optimal initializations / ensembles, (iii) it allows novelty tests against nearest-neighbor retrieval, aligned interpolation, and model soups, and (iv) it provides a clean intermediate regime before richer conditional settings, where the utility is clearer.
>
> We will sharpen Sec. 3.5 so the evaluation criteria for this regime are explicit, and we will state more clearly that fixed-architecture/task generation is a scientifically useful and already practical intermediate regime, but not the endpoint of the position. We discuss other regimes, e.g., varying architecture/task, in our response to Reviewer MMf1.
>
> Your comment on “despite extreme overparameterization” is well taken. Our point is that connectivity is striking given the nominal dimensionality, but in practice it is partly enabled by overparameterization. We will revise that sentence. We also appreciate the typo catches and will fix them.

---

> > ### Author Rebuttal · Reviewer_FGfh · 2026-04-03
> >
> > Fully resolved, thanks for your careful consideration.

---

### Decision · Program_Chairs · 2026-04-30

**Decision:**

Accept (regular)

**Comment:**

The position of this paper is that "neural network checkpoints constitute a first-class data modality, and that generative modeling in weight space should be standardized as a core machine learning primitive".

The reviewers are all positive about this paper. They believe that
- "the paper is exceptionally well written and articulated" (Reviewer FGfh)
- "the problem is feasible, has a potential to unlock much more efficient solutions for a range of problems, yet challenging" (Reviewer FGfh)
- "presents a unified framework that organizes the current progress in this area while also highlighting key unresolved problems" (Reviewer MMf1)
- "The paper is timely, only in recent years have model checkpoints been made publicly available at scale on platforms such as HF and modelscope" (Reviewer j5ih)
- "the topic is relevant to the ICML community and could potentially inspire discussion" (Reviewer prJ2)

There are some limitations to the paper as well. For instance,
- "The paper takes a relatively long time to introduce the main point." (Reviewer MMf1)
- "The current presentation remains relatively high-level and may be difficult for readers to grasp intuitively." (Reviewer MMf1)
- It doesn't provide an in-depth analysis of the fundamental obstacles. (Reviewer prJ2)
- "The discussion of alternative viewpoints is relatively shallow." (Reviewer prJ2)
- "The manuscript does not clearly identify the most important unsolved technical problems to make the proposed position work". (Reviewer prJ2)

It seems that most of these issues are easily fixable and the reviewers are generally satisfied with the paper and the suggested modification. The final recommendations for this paper are: 2x Accept and 2x Borderline Accept. The reviewers also believe that the paper has a Discussion Potential (2x Good, 2x Excellent). Therefore, I recommend the acceptance of this paper.